# From Inference to Generation: End-to-end Fully Self-supervised Generation of Human Face from Speech

*Hyeong-Seok Choi[1], *Changdae Park[2], Kyogu Lee[1]
[1]Music and Audio Research Group (MARG), Seoul National University
[2]Department of Computer Science and Engineering, Hong Kong University of Science and Technology
kekepa15@snu.ac.kr, cdpark@connect.ust.hk, kglee@snu.ac.kr

## Abstract

This work seeks the possibility of generating the human face from voice solely based on the audio-visual data without any human-labeled annotations. To this end, we propose a multi-modal learning framework that links the inference stage and generation stage. First, the inference networks are trained to match the speaker identity between the two different modalities. Then the trained inference networks cooperate with the generation network by giving conditional information about the voice. The proposed method exploits the recent development of GANs techniques and generates the human face directly from the speech waveform making our system fully end-to-end. We analyze the extent to which the network can naturally disentangle two latent factors that contribute to the generation of a face image - one that comes directly from a speech signal and the other that is not related to it - and explore whether the network can learn to generate natural human face image distribution by modeling these factors. Experimental results show that the proposed network can not only match the relationship between the human face and speech, but can also generate the high-quality human face sample conditioned on its speech. Finally, the correlation between the generated face and the corresponding speech is quantitatively measured to analyze the relationship between the two modalities.

## 1 Introduction

Utilizing audio-visual cues together to recognize a person's identity has been studied in various fields from neuroscience (Hasan et al., 2016; Tsantani et al., 2019) to practical machine learning applications (Nagrani et al., 2018b;a; Wen et al., 2019a; Shon et al., 2019). For example, some neurological studies have found that in some cortical areas, humans recognize familiar individuals by combining signals from several modalities, such as faces and voices (Hasan et al., 2016). In conjunction with the neurological studies, it is also a well known fact that a human speech production system is directly related to the shape of the vocal tract (Mermelstein, 1967; Teager & Teager, 1990).

Inspired by the aforementioned scientific evidence, we would like to ask three related questions from the perspective of machine learning: 1) Is it possible to match the identity of faces and voices? (inference) 2) If so, is it possible to generate a face image from a speech signal? (generation) 3) Can we find the relationship between the two modalities only using cross-modal self-supervision with the data "in-the-wild"? To answer these questions, we design a two-step approach where the inference and generation stages are trained sequentially. First, the two inference networks for each modality (speech encoder and face encoder) are trained to extract the useful features and to compute the cross-modal identity matching probability. Then the trained inference networks are transferred to the generation stage to pass the information about the speech, which helps the generation network to output the face image from the conditioned speech.

We believe, however, that it is impossible to perfectly reconstruct all the attributes in the image of a person's face through the characteristics of the voice alone. This is due to factors that are clearly

---

*Equal contribution

unrelated to one's voice, such as lighting, glasses, and orientation, that also exist in the natural face image. To reflect the diverse characteristics presented in the face images "in-the-wild", we therefore model the generation process by incorporating two latent factors into the neural network. More specifically, we adopted conditional generative adversarial networks (cGANs) (Mirza & Osindero, 2014; Miyato & Koyama, 2018) so that the generator network can produce a face image that is dependent not only on the paired speech condition, but also on the stochastic variable. This allows the latent factors that contribute to the overall facial attributes to be disentangled into two factors: one that is relevant to the voice and the other that is irrelevant.

Adopting cGANs negligently still leaves a few problems. For example, the condition in a cGANs framework is typically provided as embedded conditional vectors through the embedding look-up table for one-hot encoded labels (Brock et al., 2019; Miyato & Koyama, 2018). The raw signals such as speech, however, cannot be taken directly from the embedding look-up table, so an encoder module is required. Therefore, the trained speech encoder from the inference step is reused to output a pseudo conditional label that is used to extract meaningful information relevant to the corresponding face. Then the generator and the discriminator are trained in an adversarial way by utilizing the pseudo-embedded conditional vectors obtained from the trained speech encoder in the first step.

Another problem with applying the conventional cGANs for generating faces from voice arises from the fact that the distinction between different speakers can be quite subtle, which calls for a need for a more effective conditioning method. To mitigate this problem, we propose a new loss function, relativistic identity cGANs (relidGANs) loss, with modification of the relativistic GANs (Jolicoeur-Martineau, 2019), allowing us to generate the face with a more distinct identity.

Each step will be described in greater detail in Section 3.

Our contributions can be summarized as follows:

1. We propose simple but effective end-to-end inference networks trained on audio-visual data without any labels in a self-supervised manner that perform a cross-modal identity matching task.

2. A cGANs-based generation framework is proposed to generate the face from speech, to be seamlessly integrated with the trained networks from inference stage.

3. A new loss function, so called a relidGANs loss, is designed to preserve a more consistent identity between the voices and the generated images.

4. An extensive analysis is conducted on both inference and generation tasks to validate our proposed approaches.

## 2 RELATED WORKS

There has been an increasing interest in the self-supervised learning framework within the machine learning research community. While the focus of the many studies has been concentrated on applications of matching the audio-visual correspondence such as the presence or absence of objects in a video or a temporal alignment between two modalities (Chung et al., 2017; Afouras et al., 2018b;a; Ephrat et al., 2018), growing attention has come to matching the speaker *identity* between the human face and voice.

**Inference** The cross-modal identity matching task between face and voice has been recently studied in machine learning community. Nagrani et al. (2018b) proposed a convolutional-neural-network (CNN) based biometric matching network by concatenating the two embedding vectors from each modality and classifying them with an additional classifier network. Though showing promising results, this style of training is limited as the model is not flexible in that the number of concatenated vectors used in the training cannot be changed in the test phase. Next, Nagrani et al. (2018a) modeled the concept of personal identity nodes (Bruce & Young, 1986) by mapping each modality into the shared embedding space and using triplet loss to train the encoders to allow more flexible inference. Lastly, Wen et al. (2019a) proposed DIMNet where two independent encoders are trained to map the speech and face into the shared embedding space, classifying each of them independently utilizing the supervised learning signals from labels such as identity, gender and nationality.

**Generation** There have been a few concurrent works that tackled the similar problem of generating a face image from speech, which we think are worth noting here. Duarte et al. (2019) proposed a GANs-based framework to generate a face image from the speech. But the intention of their work was not to generate a face from unseen speaker identity but more of seeking the possibility of cross-modal generation between the speech and face itself. Oh et al. (2019) proposed a similar work called speech2face. The pre-trained face recognition network and the neural parametric decoder were used to produce a normalized face (Parkhi et al., 2015; Cole et al., 2017). After that, the speech encoder was trained to estimate the input parameter of the face decoder directly. Because the training process still requires the pre-trained modules trained with supervision, we do not consider this a fully-self-supervised approach. Lastly, the most similar work to ours was recently proposed by Wen et al. (2019b) where they utilized a pre-trained speech identification network as a speech encoder, and used a GANs-based approach to produce the face conditioned on the speech embedding.

Our proposed method differs from the abovementioned approaches in the following ways. First, none of the previous approaches has tried to model the stochasticity in the generation process, but we address this problem by incorporating the stochasticity in the latent space so that the different face images can be sampled even when the speech condition is fixed. Second, modeling the image is more challenging in our work as we aim to train our network to produce larger image size (128 × 128) compared to other GANs-based works. Also, we trained the model on the AVSpeech dataset (Ephrat et al., 2018) which includes extremely diverse dynamics in the images. Finally, the important scope of this work is to seek the possibility of training the whole inference and generation stages only using the self-supervised learning method, which is the first attempt to our knowledge. The whole pipeline of the proposed approach is illustrated in Fig. 1.

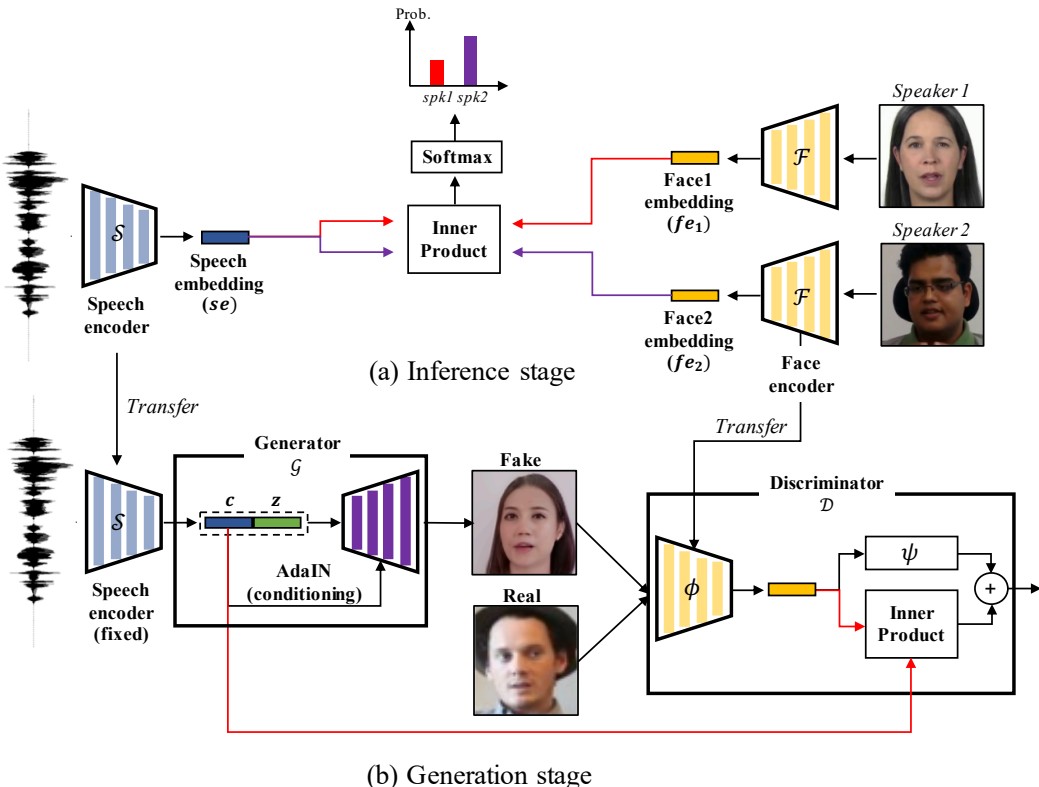

Figure 1: Overview of the proposed inference and generation stages.

## 3 SELF-SUPERVISED INFERENCE AND GENERATION

### 3.1 CROSS-MODAL IDENTITY MATCHING

In order to successfully identify the speaker identity from the two audio-visual modalities, we trained two encoders for each modality. First, a speech encoder is trained to extract the information related to a person's identity from a segment of speech. To this end, we use raw-waveform based speech encoder that was shown to extract useful features in speaker verification task, outperforming conventional feature such as mel-frequnecy-cepstal-coefficient (MFCC) even when trained using self-supervised learning in speech modality (Pascual et al., 2019)[1]. The first layer of the speech encoder is SincNet where the parameterized sinc functions act as trainable band-pass filters (Ravanelli & Bengio, 2018). The rest of the layers are composed of 7 stacks of 1d-CNN, batch normalization (BN), and multi-parametric rectified linear unit (PReLU) activation. Next, we used a 2d-CNN based face encoder with residual connections in each layer. Note that the network structure is similar to the discriminator network of (Miyato & Koyama, 2018). The details of the networks are shown in Appendix C.

Based on the speech encoder $\mathcal{S}(\cdot)$ and the face encoder $\mathcal{F}(\cdot)$, they are trained to correctly identify whether the given face and speech is paired or not. Specifically, as a cross-modal identity matching task, we consider two settings as in (Nagrani et al., 2018b), 1. V-F: given a segment of speech select one face out of $K$ different faces, 2. F-V: given an image of face select one speech segment out of $K$ different speech segments. The probability that the $j$-th face $\boldsymbol{f}_j$ is matched to the given speech $\boldsymbol{s}$ or vice versa is computed by the inner product of embedding vectors from each module followed by the softmax function as follows:

$$
\begin{aligned}
&1.\, V\text{-}F : p(y = j|\{\mathcal{S}(\boldsymbol{s}), \mathcal{F}(\boldsymbol{f}_j)\}_{j=1}^{j=K}) = p(y = j|\{\boldsymbol{se}, \boldsymbol{fe}_j\}_{j=1}^{j=K}) = \frac{e^{<\boldsymbol{se}, \boldsymbol{fe}_j>}}{\sum_{j=1}^{j=K} e^{<\boldsymbol{se}, \boldsymbol{fe}_j>}}, \\
&2.\, F\text{-}V : p(y = j|\{\mathcal{S}(\boldsymbol{s}_j), \mathcal{F}(\boldsymbol{f})\}_{j=1}^{j=K}) = p(y = j|\{\boldsymbol{se}_j, \boldsymbol{fe}\}_{j=1}^{j=K}) = \frac{e^{<\boldsymbol{se}_j, \boldsymbol{fe}>}}{\sum_{j=1}^{j=K} e^{<\boldsymbol{se}_j, \boldsymbol{fe}>}},
\end{aligned}
\tag{1}
$$

where $\boldsymbol{s}$ denotes a speech segment, $\boldsymbol{f}$ denotes a face image, $\boldsymbol{se}$ denotes a speech embedding vector and $\boldsymbol{fe}$ denotes a face embedding vector.

Note that using inner product as a means of computing similarity allows more flexible inference. For example, the proposed method enables both F-V and V-F settings, no matter with which method the model is trained. Also, it allows setting a different number of $K$ in the test phase than the training phase.

Finally, the two encoders are trained solely using the self-supervised learning signal from cross-entropy error.

### 3.2 GENERATING FACE FROM VOICE

Although we aim to generate the human face from the speech, it is only natural to think that not all attributes of the face image are correlated to the speech. Hence, we assume the latent space of the face to be broken down into two parts, the deterministic variable from the speech encoder $\boldsymbol{c}$ and a random variable $\boldsymbol{z}$ sampled from Gaussian distribution.

Such latent space is modeled using cGANs, a generative model that allows the sampling of face images conditioned on speech. More specifically, randomly sampled Gaussian noise $\boldsymbol{z} \in \mathbb{R}^{128}$ and speech condition $\boldsymbol{c} \in \mathbb{R}^{128}$ are concatenated and used as input for the generator function $\boldsymbol{f}^{fake} = \mathcal{G}(\boldsymbol{z}, \boldsymbol{c})$ to sample the face image conditioned on the speech. In addition, adaptive instance normalization (AdaIN) technique is applied as a more direct conditioning method for each layer of the generator network (Huang & Belongie, 2017; Karras et al., 2019). The details of the network are described in Appendix C.

In order to generate the face image that are not only close enough to the real face image distribution, but matches with the given condition, the condition information must be properly fed into the discriminator. Many studies have suggested such conditioning approaches for the discriminator (Reed

---

[1]The speech encoder can be downloaded in the following link: https://github.com/santi-pdp/pase

et al., 2016; Odena et al., 2017; Miyato & Koyama, 2018); among them, we adopted a recent conditioning method, projection discriminator (Miyato & Koyama, 2018), which not only suggests a more principled way of conditioning embeddings into the discriminator, but has also been widely used in many successful GANs related works (Brock et al., 2019; Zhang et al., 2018; Miyato et al., 2018). The study showed that, writing the discriminator function as $\mathcal{D}(\boldsymbol{f}, \boldsymbol{c}) := \mathcal{A}(g(\boldsymbol{f}, \boldsymbol{c}))$, the condition information can be effectively provided to the discriminator using the inner-product of two vectors, $\boldsymbol{c}$ and $\phi(\boldsymbol{f})$, as follows:

$$g(\boldsymbol{f}, \boldsymbol{c}) = \boldsymbol{c}^T \phi(\boldsymbol{f}) + \psi(\phi(\boldsymbol{f})), \tag{2}$$

where $\mathcal{A}(\cdot)$ denotes an activation function (sigmoid in our case), $\boldsymbol{c}$ denotes condition embedding, $\phi(\boldsymbol{f})$ denotes output from the inner layer of discriminator and $\psi(\cdot)$ denotes a function that maps input vector to a scalar value (fully-connected layer in our case). Here we focused on the fact that the conditioning signals can be used as an inner-product of two vectors and replaced it with the same inner-product operation used to compute the identity matching probability in the inference framework. Accordingly, we can rewrite the Eq. 2 by providing the condition with the trained speech encoder $\boldsymbol{c} = \mathcal{S}(\boldsymbol{s})$ and substituting $\phi(\cdot)$ with a trained face encoder $\mathcal{F}(\cdot)$ from the subsection 3.1 as follows:

$$g(\boldsymbol{f}, \boldsymbol{c}) = \boldsymbol{c}^T \phi(\boldsymbol{f}) + \psi(\phi(\boldsymbol{f})) = \mathcal{S}(\boldsymbol{s})^T \mathcal{F}(\boldsymbol{f}) + \psi(\mathcal{F}(\boldsymbol{f})). \tag{3}$$

Next, we adopted relativistic GANs (relGANs) loss which was reported to give stable image generation performance. See Jolicoeur-Martineau (2019) for more details. Combining the condition term $\boldsymbol{c}^T \phi(\boldsymbol{f})$ and relGANs loss, $g(\boldsymbol{f}, \boldsymbol{c})$ can be modified to $g^{rel}(\boldsymbol{f}^{real}, \boldsymbol{f}^{fake}, \boldsymbol{c})$ as follows:

$$
\begin{aligned}
g^{rel}(\boldsymbol{f}^{real}, \boldsymbol{f}^{fake}, \boldsymbol{c}) &= g(\boldsymbol{f}^{real}, \boldsymbol{c}) - g(\boldsymbol{f}^{fake}, \boldsymbol{c}) \\
&= \boldsymbol{c}^T \phi(\boldsymbol{f}^{real}) - \boldsymbol{c}^T \phi(\boldsymbol{f}^{fake}) + \psi(\phi(\boldsymbol{f}^{real})) - \psi(\phi(\boldsymbol{f}^{fake})), \\
g^{rel}(\boldsymbol{f}^{fake}, \boldsymbol{f}^{real}, \boldsymbol{c}) &= g(\boldsymbol{f}^{fake}, \boldsymbol{c}) - g(\boldsymbol{f}^{real}, \boldsymbol{c}) \\
&= \boldsymbol{c}^T \phi(\boldsymbol{f}^{fake}) - \boldsymbol{c}^T \phi(\boldsymbol{f}^{real}) + \psi(\phi(\boldsymbol{f}^{fake})) - \psi(\phi(\boldsymbol{f}^{real})).
\end{aligned}
\tag{4}
$$

Eq. 4 is formulated to produce a face from the paired speech, but it can cause catastrophic forgetting on the trained $\phi(\cdot)$ because the discriminator is no longer trained to penalize the mismatched face and voice. Thus, we again modify Eq. 4 so that the discriminator relativistically penalizes the mismatched face and voice more than a positively paired face and voice as follows:

$$
\begin{aligned}
g^{relid}(\boldsymbol{f}^{real}, \boldsymbol{f}^{fake}, \boldsymbol{c}_+, \boldsymbol{c}_-) &= g^{rel}(\boldsymbol{f}^{real}, \boldsymbol{f}^{fake}, \boldsymbol{c}_+) + \boldsymbol{c}_+^T \phi(\boldsymbol{f}^{real}) - \boldsymbol{c}_-^T \phi(\boldsymbol{f}^{real}), \\
g^{relid}(\boldsymbol{f}^{fake}, \boldsymbol{f}^{real}, \boldsymbol{c}_+, \boldsymbol{c}_-) &= g^{rel}(\boldsymbol{f}^{fake}, \boldsymbol{f}^{real}, \boldsymbol{c}_+) + \boldsymbol{c}_+^T \phi(\boldsymbol{f}^{fake}) - \boldsymbol{c}_-^T \phi(\boldsymbol{f}^{fake}),
\end{aligned}
\tag{5}
$$

where $\boldsymbol{f}^{real}$ and $\boldsymbol{c}_+$ denotes the paired face and speech condition from data distribution, $\boldsymbol{f}^{fake}$ denotes the generated face sample conditioned on $\boldsymbol{c}_+$, and $\boldsymbol{c}_-$ denotes the speech condition with mismatched identity to $\boldsymbol{f}^{real}$ using negative sampling.

Finally, utilizing the non-saturating loss (Goodfellow et al., 2014; Jolicoeur-Martineau, 2019), the proposed objective function (relidGANs loss) for discriminator $L_{\mathcal{D}}$ and generator $L_{\mathcal{G}}$ becomes as follows:

$$
\begin{aligned}
L_{\mathcal{D}} &= -\mathbb{E}_{(\boldsymbol{f}^{real}, \boldsymbol{c}_+, \boldsymbol{c}_-) \sim p_{data}, \boldsymbol{f}^{fake} \sim p_{gen}}[log(\mathcal{A}(g^{relid}(\boldsymbol{f}^{real}, \boldsymbol{f}^{fake}, \boldsymbol{c}_+, \boldsymbol{c}_-)))], \\
L_{\mathcal{G}} &= -\mathbb{E}_{(\boldsymbol{f}^{real}, \boldsymbol{c}_+, \boldsymbol{c}_-) \sim p_{data}, \boldsymbol{f}^{fake} \sim p_{gen}}[log(\mathcal{A}(g^{relid}(\boldsymbol{f}^{fake}, \boldsymbol{f}^{real}, \boldsymbol{c}_+, \boldsymbol{c}_-)))].
\end{aligned}
\tag{6}
$$

## 4 EXPERIMENTS AND RESULTS

**Dataset and Sampling** Two datasets were used throughout the experiments. The first dataset is the AVSpeech dataset (Ephrat et al., 2018). It consists of 2.8m of YouTube video clips of people actively speaking. Among them, we downloaded about 800k video clips from a train set and 140k clips from a test set. Out of 800k training samples, we used 50k of them as a validation set. The face images included in the dataset are extremely diverse since the face images were extracted from the videos "in the wild" (e.g., closing eyes, moving lips, diversity of video quality, and diverse facial expressions) making it challenging to train a generation model. In addition, since no speaker identity information is provided, training the model with this dataset is considered *fully self-supervised*. Because of the absence of the speaker identity information, we assumed that each audio-visual clip represents an

individual identity. Therefore, a positive audio-visual pair was sampled only within a single clip while the negative samples were randomly selected from any clips excluding the ones sampled by the positive pair. Note that each speech and face image included in the positive pair was sampled from different time frames within the clip. This was to ensure that the encoded embeddings of each modality do not contain linguistic related features (e.g., lip movement or phoneme related features).

The second dataset is the intersection of VoxCeleb (Nagrani & Zisserman, 2017) and VGGFace (Parkhi et al., 2015). VoxCeleb is also a large-scale audio-visual dataset collected on YouTube videos. VGGFace is a collection of the images of public figures gathered from the web, which is less diverse and relatively easier to model compared to the images in AVSpeech dataset. Since both VGGFace and VoxCeleb provide speaker identity information, we used the speakers' included in the both datasets resulting in 1,251 speakers. We used face images from both VoxCeleb and VGGFace, and used speech audio from Voxceleb. Note that, this dataset provides multiple images and audio for a single speaker; therefore, training a model with this dataset cannot be called a self-supervised training method in a strict sense.

**Implementation** In every experiment, we used 6 seconds of audio. The speech samples that were shorter or longer than 6 seconds were duplicated or randomly truncated so that they became 6 seconds. At the inference stage, we trained the networks with stochastic gradient descent optimizer (SGD). At the generation stage, we trained the networks with Adam optimizer (Kingma & Ba, 2015). The discriminator network was regularized using $R_1$ regularizer (Mescheder et al., 2018). More details are described in Appendix B.

## 4.1 Cross-modal Inference Evaluation Results

**Fully self-supervised inference accuracy results**: Here we show the evaluation results of the inference networks trained on the AVSpeech dataset. Again, the model was trained with no additional information about speaker identity. The evaluations were conducted 5 times by selecting different negative samples each time and reported using the average value of them.

We trained our model in two settings (1. V-F and 2. F-V) and both were trained in 10-way setting (1 positive pair and 9 negative pairs). Then the two trained models were tested on both V-F and F-V settings, each of which were tested on 2-way and 10-way settings. The top1-accuracy (%) results of cross-modal identity matching task is shown in Table 1. The results show that our network can perform the task of cross-modal identity matching with a reasonable accuracy despite being trained in a fully self-supervised manner. In addition, our model can perform F-V inference with a reasonable accuracy even when it is trained in V-F setting, and vice versa.

Table 1: Accuracy (%) of cross-modal identity matching task.

| Test \ Train | V-F | | F-V | |
|---|---|---|---|---|
| V-F | **89.22** | **54.33** | 85.10 | 44.70 |
| F-V | 86.94 | 49.20 | **88.47** | **52.55** |
| *K*-way | 2 | 10 | 2 | 10 |

**Comparison results**: We compared our model with two models - SVHFNet (Nagrani et al., 2018b) and DIMNet (Wen et al., 2019a). For fair comparisons, we trained our model with the intersection of VoxCeleb and VGGFace datasets. Our model is trained in two train settings (1. V-F and 2. F-V) with 10-way configuration.

The comparison results are shown in Table 2. The results show that our model performs worse when tested in 2-way setting compared to other models. Note that, SVHFNet used a pre-trained speaker identification network and face recognition network trained in a supervised manner with identity information and DIMNet trained the network in a supervised manner using the labels such as identity, gender, and nationality, and therefore is not trainable without such labels, whereas our model was trained from the scratch and without any such labels. Under a more challenging situation where $K$ increases from two to ten, however, our model shows significantly bet-

Table 2: Comparison results in terms of accuracy (%) of cross-modal identity matching task.

| Model \ Train | V-F [2] | | F-V | |
|---|---|---|---|---|
| SVHFNet | 81.00 | 35 | 79.50 | - |
| DIMNet-IG | **84.12** | 40 | **84.03** | - |
| Ours | 79.90 | **55.66** | 80.83 | **54.84** |
| *K*-way | 2 | 10 | 2 | 10 |

---

[2]The 10-way results are inferred from the graph of the paper DIMNet (Wen et al., 2019a).

ter performance, which may indicate the proposed method is capable of extracting more informative and reliable features for the cross-modal identity matching task.

## 4.2 CROSS-MODAL GENERATION EVALUATION RESULTS

We conducted three qualitative analyses (QLA's) and three quantitative analyses (QTA's) to thoroughly examine the relationship between condition and the generated face samples.

**QLA 1. Random samples from $(z, c)$ plane**: The generated face images from diversely sampled $z$ and fixed speech condition $c$ is shown in Fig. 2. We can observe that each variable shows different characteristics. For example, it is observable that $z$ controls the orientation of head, background color, haircolor, hairstyle, glasses, etc. Alternatively, we observed that $c$ controls gender, age, ethnicity and the details of the face such as the shape of the face and the shape of the nose.

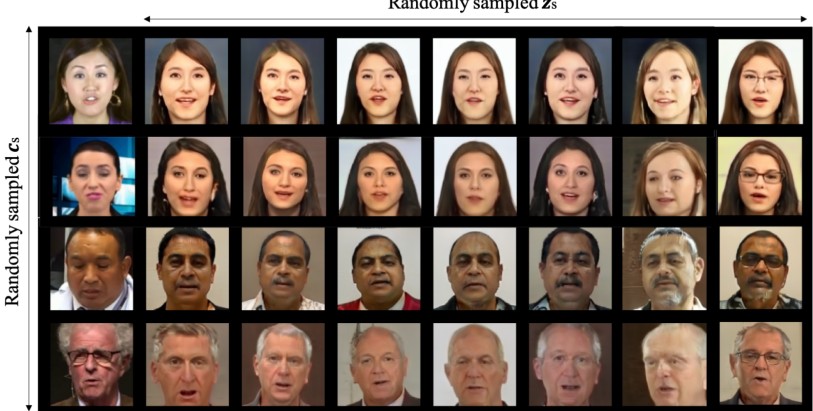

Figure 2: The illustration of generated face images from interpolated speech conditions. Note that the first column consists of the ground truth image of the speakers[3].

**QLA 2. Generated samples from interpolated speech condition**: Next, we conducted an experiment to observe how speech conditions affect facial attributes by exploring the latent variable $c$ with $z$ fixed. We sampled the speech segments from the AVSpeech test set and linearly interpolated two speech conditions. After that we generated the face images based on the interpolated condition vectors. The generated face images are shown in Fig. 3. We can observe the smooth transition of speech condition $c$ in the latent space and can confirm that it has a control over the parts that are directly correlated to the identity of a person such as gender, age, and ethnicity.

**QLA 3. Generated samples from interpolated stochastic variable**: Lastly, we conducted an experiment that shows the role of stochastic variable in the generation process by interpolating two randomly sampled $z$ vectors with $c$ fixed. The speech segments from the AVSpeech test set were used to make a fixed $c$ for each row. The generated face images are shown in Fig. 4. We can observe that the latent vector $z$ has a control over the parts that are not directly correlated to the identity of a person such as head orientation, hair color, attire, and glasses.

**QTA 1. Correlation between the generated samples and speech conditions**: We performed a quantitative analysis to investigate the relationship between the speech condition and the face image it generated. To this end, we first sampled two different random variables $z_1$, $z_2$ and two different speech condition vectors $c_1$, $c_2$ of different speakers. Next, we generated two face images $f_1^{fake} = \mathcal{G}(z_1, c_1)$, $f_2^{fake} = \mathcal{G}(z_2, c_2)$ using the generator network. Then the two generated samples were encoded using the trained inference network $\mathcal{F}(\cdot)$ to extract the face embeddings $fe_1 = \mathcal{F}(f_1^{fake})$, $fe_2 = \mathcal{F}(f_2^{fake})$. We then calculated the cosine distance between the speech condition vectors ($CD(c_1, c_2)$), and the cosine distance between the two face embeddings ($CD(fe_1, fe_2)$). Finally, we computed the Pearson correlation between $CD(c_1, c_2)$ and $CD(fe_1, fe_2)$. This is to see if there exists any positive correlation between the embedding spaces of the two different modalities;

---

[3]Speech samples: https://drive.google.com/open?id=1n9VTYm9Z–dxNpwS-ELiotmXES6COE4h

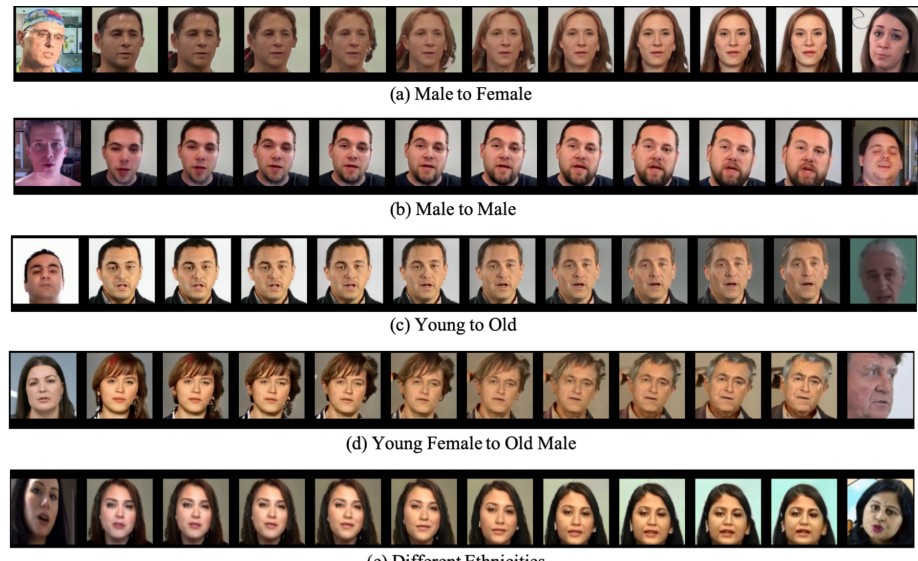

Figure 3: The illustration of generated face images from interpolated speech condition vectors while fixing $z$. Note that the images on the very left and right sides of each row are the ground truth face images of the speakers.

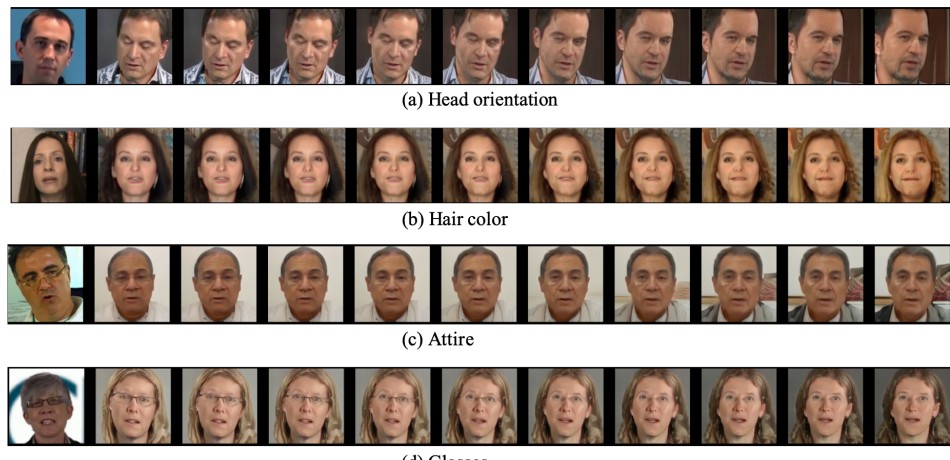

Figure 4: The illustration of generated face images from interpolated random vectors while fixing $c$. Note that the images on the very left side of each row are the ground truth face images of the speakers.

that is, to see if closer speech embeddings help generate face images whose embeddings are also closer even when the random variable $z$ is perturbed.

Fig. 5 (a) shows the results using the test set of the AVSpeech. We can see a positive correlation between the $CD(c_1, c_2)$ and $CD(\boldsymbol{fe}_1, \boldsymbol{fe}_2)$ meaning that the generated face images are not randomly sampled.

Furthermore, we examined whether the $CD$ between two speech condition vectors gets closer when controlling the gender of the speaker, and also the face images generated from the two speech condition vectors. We tested this on VoxCeleb dataset as it provides gender labels for each speaker identity. We compared the mean values of $CD(c_1, c_2)$ and $CD(\boldsymbol{fe}_1, \boldsymbol{fe}_2)$ in two cases; one setting the gender of the two speakers different and the other one setting the gender the same. Fig. 5 (b) shows a scatter plot of the $CD$ when the two sampled speakers have different genders, and Fig. 5 (c)

shows a case where the genders are set the same. We found out that the mean value of $CD(\boldsymbol{c}_1, \boldsymbol{c}_2)$ gets smaller ($0.46 \rightarrow 0.28$) when we set the gender of two speakers the same and accordingly the $CD(\boldsymbol{fe}_1, \boldsymbol{fe}_2)$ gets also smaller ($0.27 \rightarrow 0.15$).

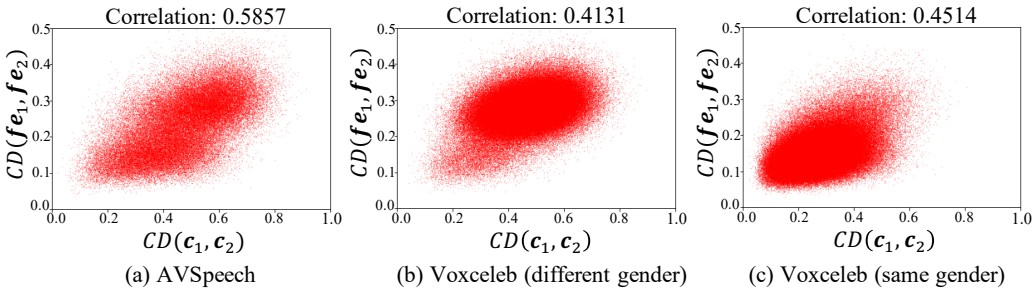

(a) AVSpeech        (b) Voxceleb (different gender)        (c) Voxceleb (same gender)

Figure 5: The scatter plots of $CD(\boldsymbol{c}_1, \boldsymbol{c}_2)$ and $CD(\boldsymbol{fe}_1, \boldsymbol{fe}_2)$.

**QTA 2. Testing the generated samples with the inference networks**: Here we conducted two inference experiments in V-F setting. In the first experiment, given a positive pair from the AVSpeech test set ($\boldsymbol{f}, \boldsymbol{s}$), two face images ($\boldsymbol{f}$ and $\mathcal{G}(\boldsymbol{z}, \mathcal{S}(\boldsymbol{s}))$) are passed to the trained inference networks. The inference networks are then used to determine which of the two images yields a greater cross-modal matching probability with a given speech segment $\boldsymbol{s}$ as follows:

$$\sum_{n=1}^{N} \mathbb{1}[p(y=1|\{\mathcal{S}(\boldsymbol{s}_n), \mathcal{F}(\boldsymbol{f}_{n,j})\}_{j=1}^{j=2}) > p(y=2|\{\mathcal{S}(\boldsymbol{s}_n), \mathcal{F}(\boldsymbol{f}_{n,j})\}_{j=1}^{j=2})]/N, \qquad (7)$$

where $\mathbb{1}[\cdot]$ denotes an identity function, $n$ denotes a index of test sample, $\boldsymbol{f}_{n,1}$ denotes the generated sample $\mathcal{G}(\boldsymbol{z}, \mathcal{S}(\boldsymbol{s}_n)))$, $\boldsymbol{f}_{n,2}$ denotes the ground truth image paired with the speech $\boldsymbol{s}_n$, and $N$ denotes the total number of test samples. Note that $\boldsymbol{z}$ was randomly sampled from Gaussian distribution for each $n$.

Surprisingly, we found out that the inference networks tend to choose the generated samples more than the ground truth face images with a chance of 76.65%, meaning that the generator is able to generate a plausible image given the speech condition $\boldsymbol{c}$. On the other hand, we found out that the chances significantly decreases to 47.2% when we do not use the relidGANs loss (using Eq. 4), showing that the proposed loss function helps generate images reflecting the identity of a speaker. Note that, however, this result does not necessarily say that the generator is capable of generating a more plausible image conditioned on one's voice than the paired face image as this experiment is bounded to the performance of the inference network. In addition, we believe the distribution of real face image is much more diverse than the generated samples, causing the inference network tend towards the generated samples more.

Our second experiment is similar to the first experiment, but this time we compared the generated images from two different generators; one trained without mismatched identity loss (Eq. 4) and the other with the proposed relidGANs loss. We found out that the inference networks selected the generated sample from the generator trained with the proposed relidGANs loss with a chance of 79.68%, again showing that the proposed loss function helps to generate samples that reflect the identity information encoded in the speech condition.

Next, we conducted one additional experiment in F-V setting. Given a generated image $\mathcal{G}(\boldsymbol{z}, \mathcal{S}(\boldsymbol{s}_1))$ from a speech segment $\boldsymbol{s}_1$, we measured the accuracy of inference network selecting $\boldsymbol{s}_1$ out of two audio segments $\boldsymbol{s}_1$ and $\boldsymbol{s}_2$ as follows:

$$\sum_{n=1}^{N} \mathbb{1}[p(y=1|\{\mathcal{S}(\boldsymbol{s}_{n,j}), \mathcal{F}(\boldsymbol{f}_n)\}_{j=1}^{j=2}) > p(y=2|\{\mathcal{S}(\boldsymbol{s}_{n,j}), \mathcal{F}(\boldsymbol{f}_n)\}_{j=1}^{j=2})]/N, \qquad (8)$$

where $\boldsymbol{f}_n$ denotes a generated face image from a speech segment $\boldsymbol{s}_{n,1}$, and $\boldsymbol{s}_{n,2}$ denotes a negatively selected speech segment. We found out that the inference networks select $\boldsymbol{s}_1$ with a chance of 95.14%. Note that, the accuracy of inference networks in 2-way F-V setting is 88.47%, which means the generator can faithfully generate a face image according to the given speech segment $\boldsymbol{s}_1$.

**QTA 3. Face image retrieval**: Lastly, we conducted a face retrieval experiment in which the goal was to accurately retrieve a real face image for the speaker using the generated image from their speech segment as a query. To compose a retrieval dataset, we randomly sampled 100 speakers from the test set of the AVSpeech. For each speaker, we extracted 50 face images from a video clip resulting in 5,000 images for the retrieval experiment in total. Note that this process of composing the retrieval dataset is same as that of Speech2Face Oh et al. (2019). To retrieve the closest face image out of the 5,000 samples, the trained face encoder was used to measure the feature distance between the generated image and each of the 5,000 images. We computed three metrics as a feature distance ($L_1$, $L_2$, and cosine distance ($CD$)) and the results are reported in Table 3.

Table 3 shows that our model can achieve higher performance than that of Speech2Face on all distance metrics. We also measured the performance of the generator trained without mismatched identity loss (mil) (Eq. 4). The results show that the proposed relidGANs loss function is crucial to generate the face images that reflect the identity of the speakers. Examples of the top-5 retrieval results are shown in Fig. 6.

Table 3: Face retrieval performance.

| Models | Metric | Top-$K$ | | | |
|---|---|---|---|---|---|
| | | $K = 1$ | $K = 2$ | $K = 5$ | $K = 10$ |
| Speech2Face | $L_1$ | 8.34 | 13.7 | 24.66 | 36.22 |
| | $L_2$ | 8.28 | 13.66 | 24.66 | 35.84 |
| | $CD$ | 10.92 | 17.00 | 30.60 | 45.82 |
| Ours (w/o mil) | $L_1$ | 7.32 | 12.81 | 24.41 | 38.82 |
| | $L_2$ | 7.21 | 12.83 | 24.34 | 39.24 |
| | $CD$ | 7.36 | 13.04 | 24.78 | 39.59 |
| Ours (w/ mil (relidGANs)) | $L_1$ | 12.97 | 20.98 | 36.56 | 52.66 |
| | $L_2$ | 12.90 | 21.5 | 36.84 | 52.49 |
| | $CD$ | **13.59** | **21.69** | **36.94** | **53.83** |

Real    Generated    Top-5 nearest face images

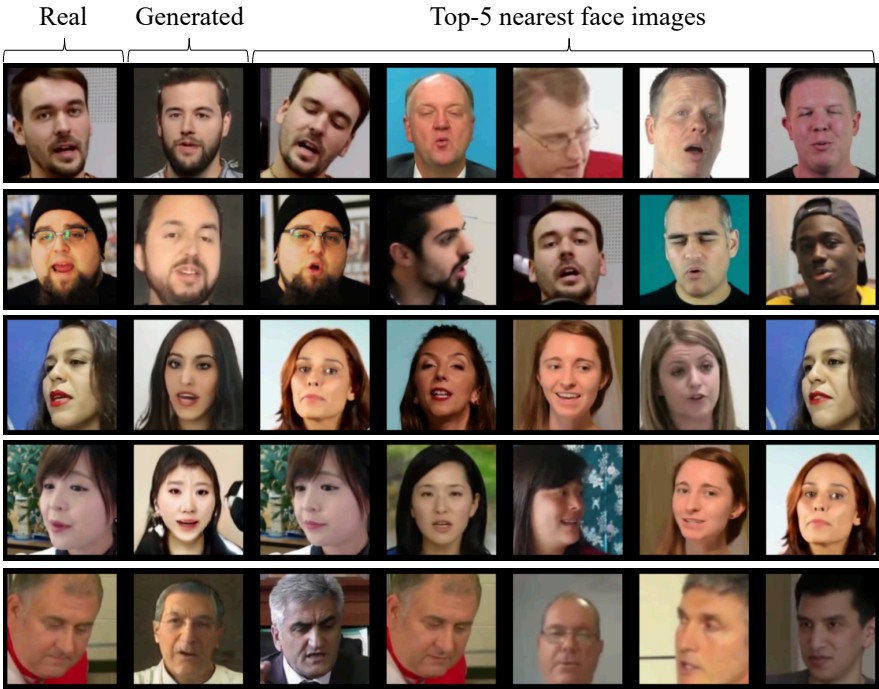

Figure 6: Top-5 retrieval examples.

## 5    CONCLUSION AND FUTURE WORK

In this work, we proposed a cross-modal inference and generation framework that can be trained in a fully self-supervised way. We trained cGANs by transferring the trained networks from the inference stage so that the speech could be successfully encoded as a pseudo conditional embedding. We also proposed relidGANs loss to train the discriminator to penalize negatively paired face and speech

so that the generator could produce face images with more distinguished identity between different speakers. As a future work, we would like to address a data bias problem (e.g., ethnicity, gender, age, etc.) that exists in many datasets. This is a significant problem as many publicly available datasets have biased demographic statistics, consequently affecting the results of many algorithms (Buolamwini & Gebru, 2018). We believe that this can be solved with the use of a better data sampling strategy in an unsupervised manner such as (Amini et al., 2019). In addition, we would like to expand the proposed methods to various multi-modal datasets by generalizing the proposed concept to other modalities.

## ACKNOWLEDGMENTS

This work was supported partly by Next-Generation Information Computing Development Program through the National Research Foundation of Korea (NRF) funded by the Ministry of Science and ICT (MSIT; Grant No. NRF-2017M3C4A7078548), and partly by Institute for Information & Communications Technology Planning & Evaluation(IITP) grant funded by the Korea government(MSIT) (No.2019-0-01367, Infant-Mimic Neurocognitive Developmental Machine Learning from Interaction Experience with Real World (BabyMind))

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

## APPENDIX

### A    DATA PREPROCESSING

**Image processing**: All video clips downloaded from the AVSpeech dataset were resampled to be 25FPS. If more than one face are detected in a frame, a face closer to the coordinates of the target speaker in the first frame was selected. Note that the coordinates of the target speakers are provided by the AVSpeech dataset. Because the size of the speaker's face on the screen varies from video to video, the image was resized to maintain interocular distance as 55 pixels, as described in Cole et al. (2017). We used a publicly available software, Dlib (King, 2009), to detect and crop the face images. The images were cropped to the size of $224 \times 224$ and resized to $128 \times 128$ for training. We additionally applied horizontal flip method as a data augmentation strategy. These procedures of cropping, resize, and flipping are adapted to all images in VGGFace and VoxCeleb, too.

**Audio processing**: All audio samples were resampled to 16kHz, and stereo audio samples were converted to mono. We used 6 seconds of audio in both the inference and generation model. If the audio is longer than 6 seconds, the audio was randomly truncated. If the audio is shorter than 6 seconds, the entire audio was duplicated until it becomes longer than 6 seconds. After then the duplicated audio sample was randomly truncated to be 6 seconds. We applied root-mean-square normalization to make the overall amplitude of speech signals to be consistent. The reference level was selected as 0.01.

### B    IMPLEMENTATION DETAILS

We used stochastic gradient descent (SGD) with the momentum of 0.9 and weight decay of $5e - 4$ to optimize inference networks (speech encoder and face encoder). The learning rate was initialized to 0.001 and decayed by the factor of 10 if validation loss was not decreased for 1 epoch. Training was stopped if learning rate decay occurred three times. Minibatch size was fixed to 32 and 12 for V-F and F-V training configuration, respectively. For both training and test phase, negative samples were randomly selected for every step while pre-defined negative samples for each positive sample were used for validation phase to ensure stable learning rate decay scheduling.

To train cGANs, we used Adam optimizer using $\beta_1$ and $\beta_1$ of 0.9 and 0.9, respectively. The learning rate of generator and discriminator was fixed to 0.0001 and 0.00005 during training, respectively. Each time the discriminator was updated twice, the generator was updated once. Batch size is 24 and we trained the model about 500,000 iterations. Note that we adopted the truncation trick of Brock et al. (2019) in subsection 4.2. In all experiments except QTA 1 and QTA 2, where the truncation trick was not adopted, the truncation threshold was set to 1.0.

## C  Network Structures

Inference network consists of a speech encoder and a face encoder. The network structure of the speech encoder is based on the problem agnostic speech encoder (PASE) (Pascual et al., 2019) followed by an additional time pooling layer and a fully-connected layer (FC). PASE consists of SincNet and 7 stacks convolutional-block (ConvBlock) composed of 1d-CNN, batch normalization, and multi-parametric rectified linear unit (PReLU) activation. Following PASE, average pooling on time dimension is applied to make the size of embedding time-invariant. The FC layer was used as the last layer of the speech encoder. The details of the speech encoder is depicted in Fig. 7 For face encoder, we used 2d-CNN based residual block (ResBlock) in each layer. Note that the network structure is similar to the discriminator network of (Miyato & Koyama, 2018). The details of the architecture is shown in Fig. 8.

For the generator, we followed the generator network structure of (Miyato & Koyama, 2018) with some modifications, such as concatenating $z$ and $c$ as an input and adopting adaptive instance normalization (AdaIN) for the direct conditioning method. The details of the discriminator are almost the same as the face encoder except that it includes additional FC layer, projection layer, and sigmoid as activation function. The details of the discriminator and generator are shown in Fig. 8 and Fig. 9, respectively.

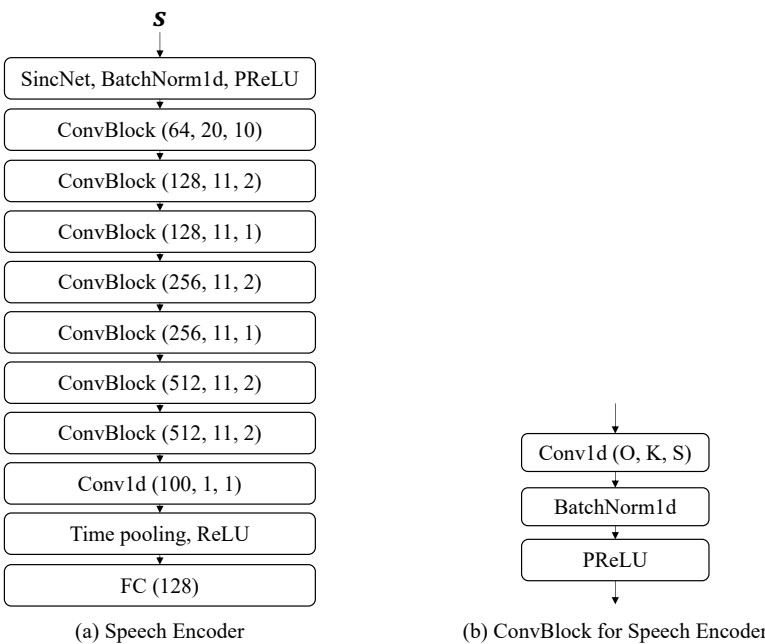

(a) Speech Encoder    (b) ConvBlock for Speech Encoder

Figure 7: The structure of the speech encoder. O, K, S indicate the number of output channels, size of the kernel and stride, respectively

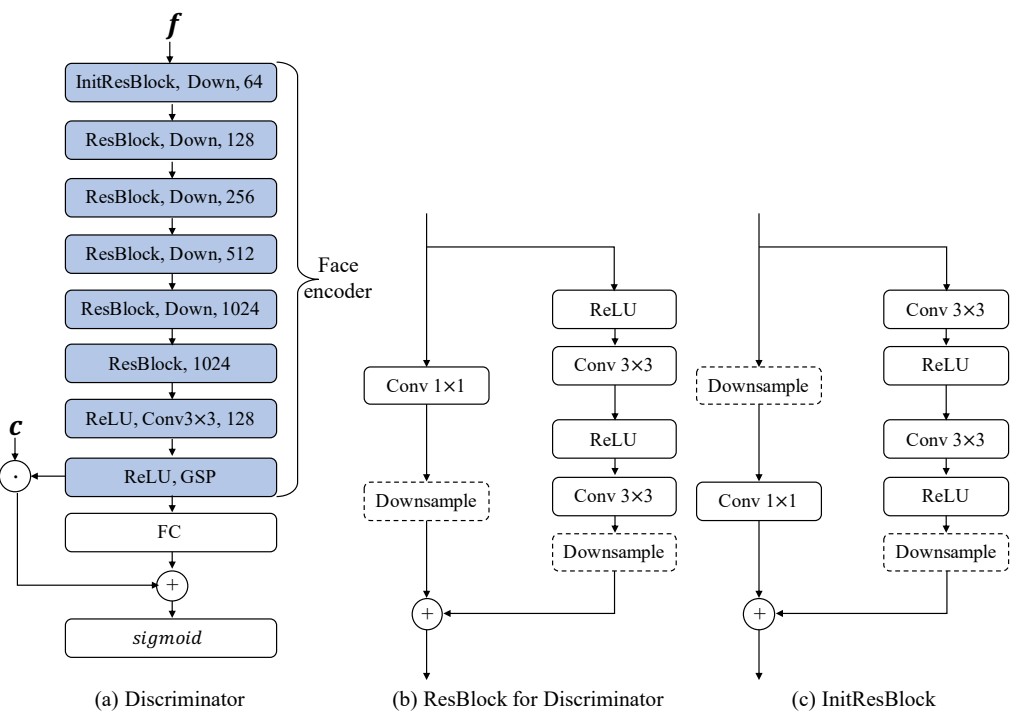

Figure 8: The structure of the discriminator network. The blue colored blocks indicate the network structure of the face encoder $\mathcal{F}$ which is transferred to the discriminator network at the generation stage. The numbers on each block denote the output channel. The GSP denotes a global sum pooling along the spatial dimension.

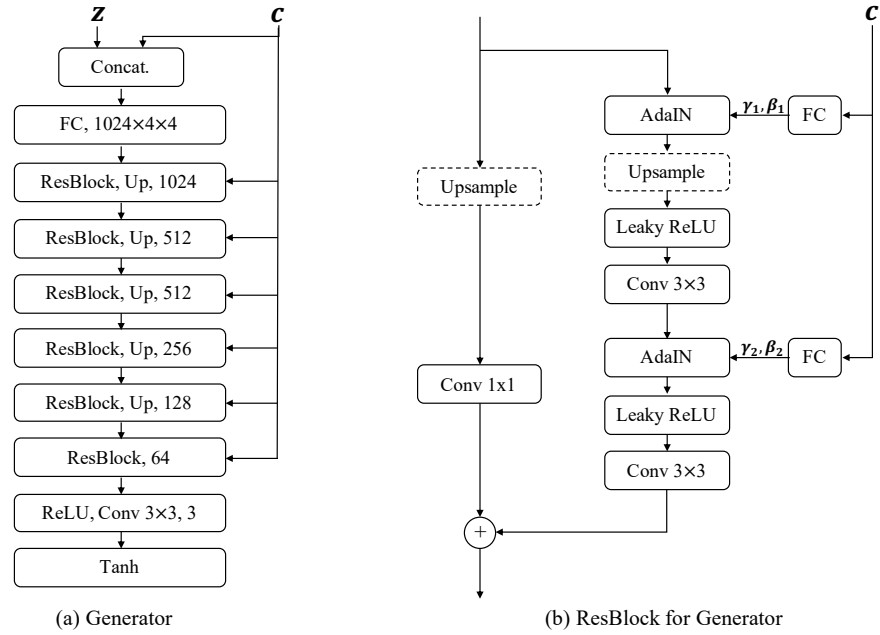

Figure 9: The structure of the generator network. The numbers on each block denote the output channel.

## D  SUPPLEMENTARY GENERATED IMAGES

Here we provide some additional uncurated generated face images. Fig. 10 and Fig. 11 show generated images with truncation threshold 0.5 and 1.0, respectively [4].

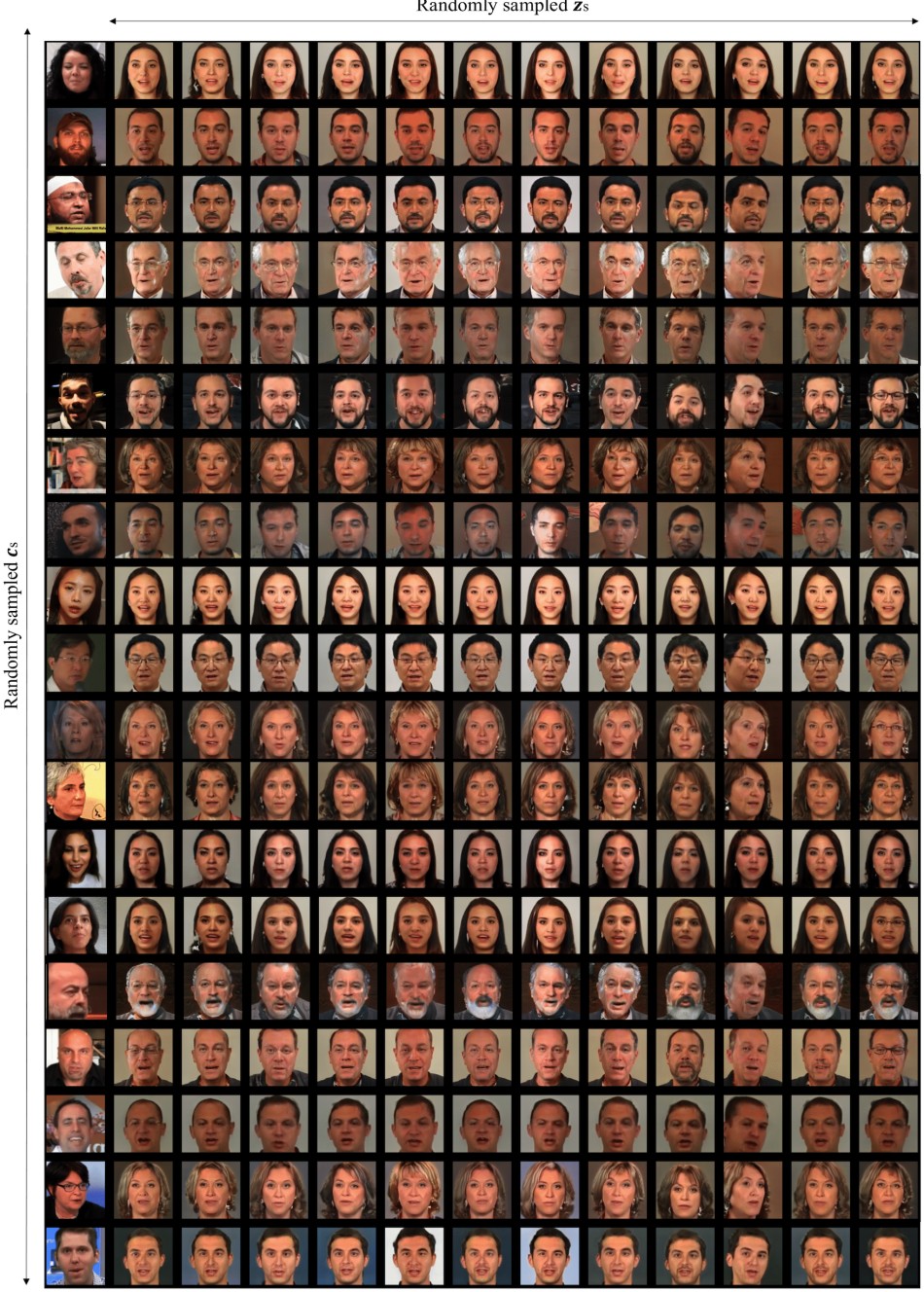

Figure 10: The uncurated generated face images of 19 speakers from the test set of the AVSpeech. The very left column is composed of the ground truth face images of speakers.

[4] Audio samples are available in the following link: https://drive.google.com/open?id=1gWZqh6VfqN33uaKDlmRf8IxmF33FoxAE

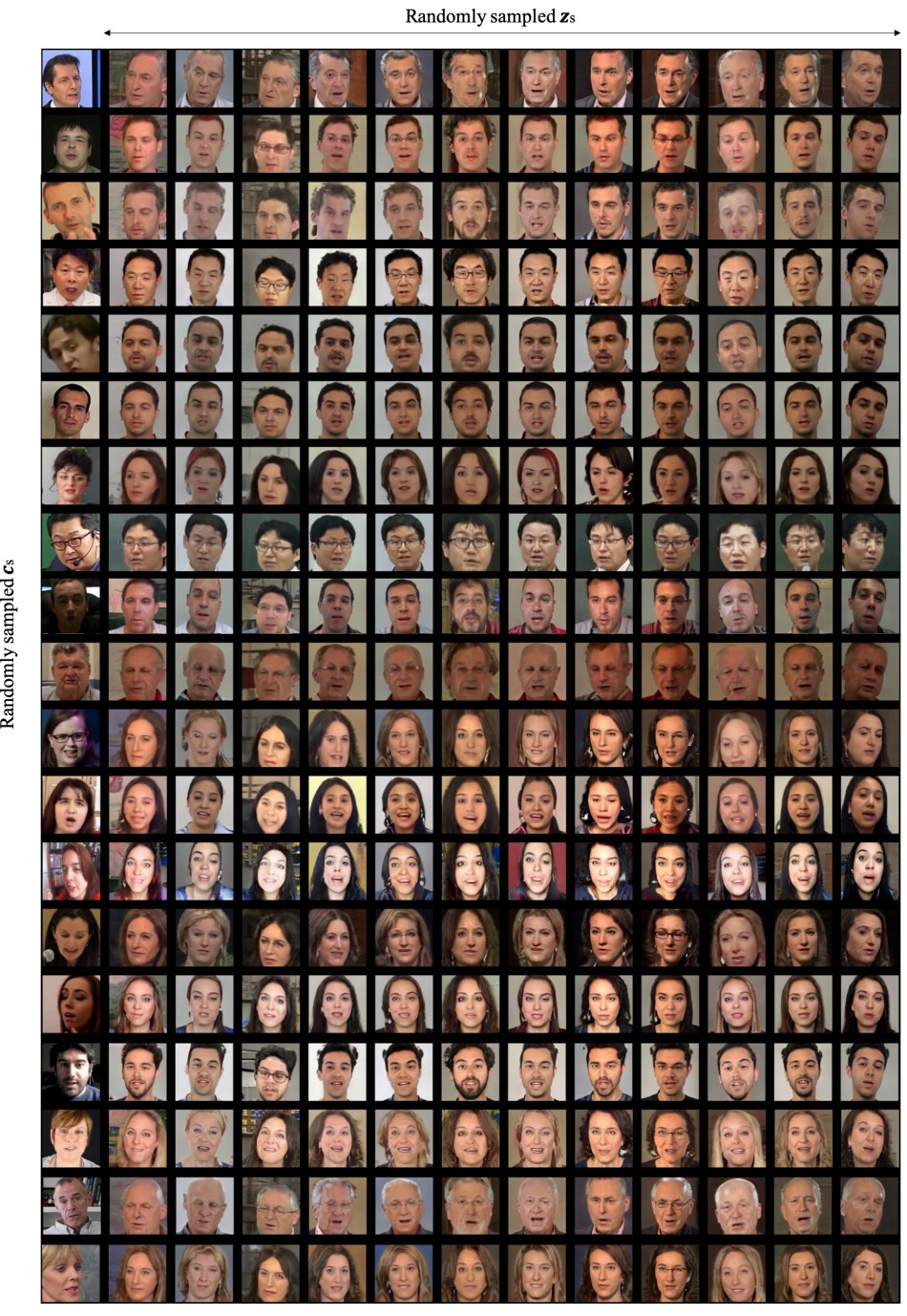

Figure 11: The uncurated generated face images of 19 speakers from the test set of the AVSpeech. The very left column is composed of the ground truth face images of speakers.

## E  THE EFFECTIVENESS OF TWO-STAGE TRAINING

In this subsection, we would like to address concerns regarding the validity of the two-stage training method as proposed in Fig. 3. Here the effect of the two-stage training method is demonstrated by showing the failure case of training when we skip the inference training stage. Fig. 12 shows that the generator ignores the speech condition vector in the generation process and, consequently, the generated images are only dependent on the random vector $z$. This shows that the learned representation using the proposed self-supervised method is effective and can be used as an important cue for the conditional generation.

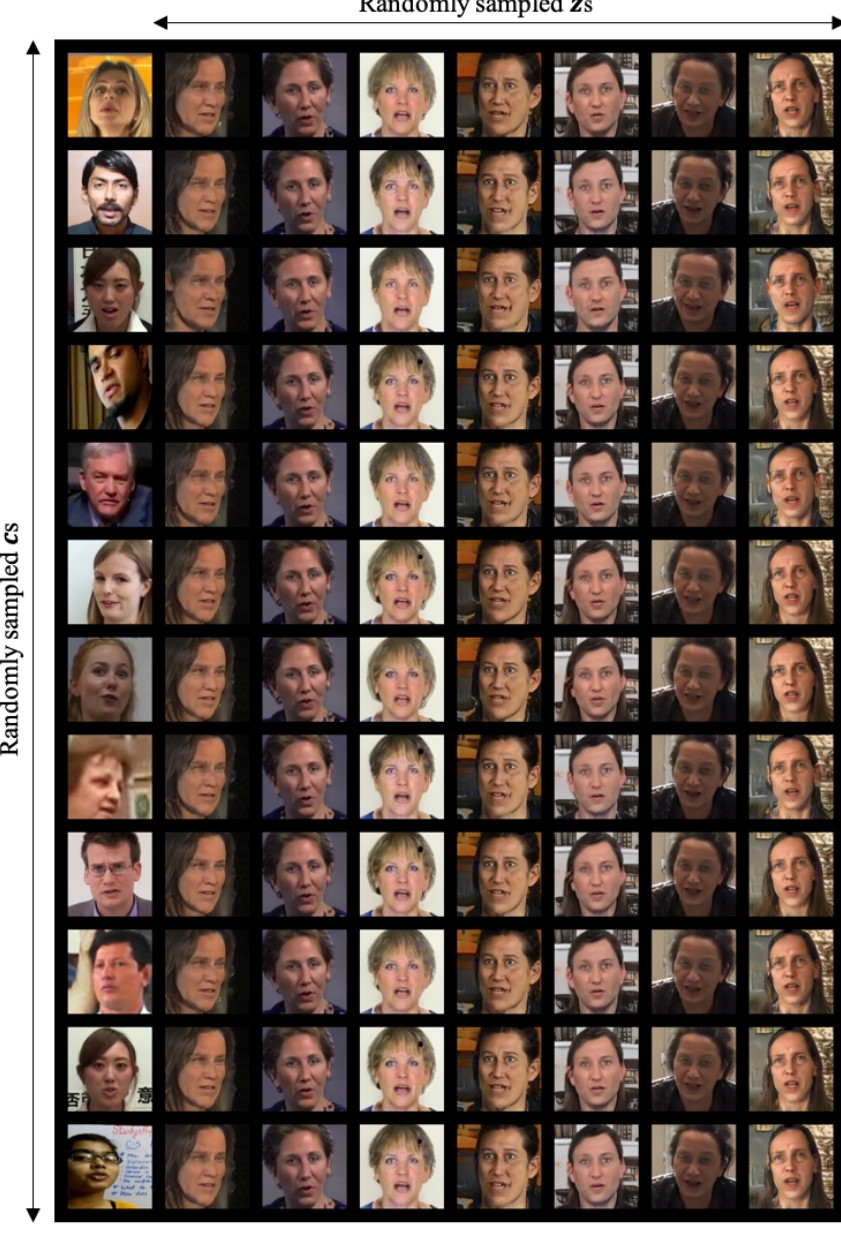

Figure 12: The generated face images when skipping the inference training stage. The very left column is composed of the ground truth face images of speakers.

