# OpenReview forum: "From Inference to Generation: End-to-end Fully Self-supervised Generation of Human Face from Speech"
_ICLR.cc/2020/Conference — Accept (Poster)_

### Official Review · AnonReviewer1 · 2019-10-22
**Official Blind Review #1**

**Rating:** 6

**Review:**

This paper presents a multi-modal learning framework that links the inference stage and generation stage for seeking the possibility of generating the human face from voice solely based on the audio-visual data without any human-labeled annotations. Experimental results show that the proposed network can not only match the relationship between the human face and speech, but can also generate the high-quality human face sample conditioned on its speech.

The writing and presentation are clear.

My concerns are as below.
1) What are the training computational complexity and testing time cost of the proposed method?
2) How can we determine truncation threshold more elegant? Any theoretical analysis and sensitive analysis?
3) How did the authors handle model collapse during training?
4) The format of references should be consistent.

**Experience Assessment:**

I have published in this field for several years.

**Review Assessment: Checking Correctness Of Derivations And Theory:**

I carefully checked the derivations and theory.

**Review Assessment: Checking Correctness Of Experiments:**

I carefully checked the experiments.

**Review Assessment: Thoroughness In Paper Reading:**

I read the paper thoroughly.

---

> ### Author Response · Authors · 2019-11-08
> **Responses to AnonReviewer1**
>
> We thank AnonReviewer1 for the careful reading on our work.
> Below are the responses to the concerns.
>
> Question 1
> What are the training computational complexity and testing time cost of the proposed method?
> Response 1
> The training took about 8 days to train the model for 16 epochs, so roughly 2 epochs for a day. We used two Titan V GPU cards with the batch size of 24.
> The generation takes about 12ms to generate a single image from a speech segment of 6-second.
>
> Question 2
> How can we determine truncation threshold more elegant? Any theoretical analysis and sensitive analysis?
> Response 2
> Unfortunately, we did not inspect thoroughly on the truncation threshold. One insight we found, however, is that as we increase the truncation threshold the generated example gets more diverse, for example, different hair styles, various hair colors, and so on.
>
> Question 3
> How did the authors handle model collapse during training?
> Response 3
> We did not encounter any mode collapse during training. A possible explanation would be attributed to the stabilized training with a R1 regularizer. We found out that without using the R1 regularizer the loss blows up and the training does not work.
>
> Question 4
> The format of references should be consistent.
> Response 4
> Thank you for pointing this out. We modified the reference format more consistently in the revised manuscript.

---

### Official Review · AnonReviewer2 · 2019-10-26
**Official Blind Review #1**

**Rating:** 3

**Review:**

This work aims to build one conditional face image generation framework from the audio signal. The whole framework is built on top of cGAN-based projector discriminator framework where the input condition vector is also used in the discriminator stage.  The authors compared with one recent work and demonstrated improvements on face retrieval experiment.


Questions:
1	In terms of differences with previous approaches,  not sure if the self-supervised learning is one of them, since it is also applied in previous Speech2Face framework.
2	Based on the listed generated examples in Fig.2, most faces are frontal, especially along Z axis, not sure if the variation of Z can determine the head orientation. Faces in the third row and the forth may not keep the identity that well, especially the comparison between the face at the first column with others.
3	One typo may need to be addressed in the first paragraph of page 5, there are “the the” before word ‘inference’.
4	As listed in Table 2, it seems the proposed approach achieves better performance when using 10 way training compared to other frameworks, any more analysis  why the proposed framework can achieve better performance in 10 way but obtain less accuracy in 2 way settings?

5	Does the test data of 2 way or 10 way experiments also include the same ratio of positive and negative pairs? If so, how about the performance on the same validation data?

6	About the results listed in Table 2, does the number 35^2 indicate 0.35?

7.    It may be necessary to include these conditional face video generation works in the related work.



**Experience Assessment:**

I have published one or two papers in this area.

**Review Assessment: Checking Correctness Of Derivations And Theory:**

I assessed the sensibility of the derivations and theory.

**Review Assessment: Checking Correctness Of Experiments:**

I carefully checked the experiments.

**Review Assessment: Thoroughness In Paper Reading:**

I read the paper at least twice and used my best judgement in assessing the paper.

---

> ### Author Response · Authors · 2019-11-08
> **Responses to AnonReviewer2**
>
> We thank AnonReviewer2 for the thorough reading and analysis on our work.
> Below are the answers to the questions.
>
> Question 1
> In terms of differences with previous approaches, not sure if the self-supervised learning is one of them, since it is also applied in previous Speech2Face framework.
> Answer 1
> Here, we have to first clarify what the “self-supervised training” is. In Speech2Face, the framework requires two pre-trained modules: 1. face recognition network and 2. face decoder.
> The face recognition network is based on the work of Parkhi et al. [1] which requires annotated identity information to train, hence is “supervised”. The face decoder network is based on the work of Cole et al. [2] and in order to train the decoder network it requires FaceNet features which were again trained using human-labeled annotated identity information.
> On the other hand, our proposed framework does not require such pre-trained modules trained in a supervised manner and that is why we tried to emphasize our work is “fully-self-supervised”, meaning that our work does not require any human-labeled annotation both in inference and generation stages which makes our work novel. We updated this part in the Related Works section of the revised manuscript.
>
> Question2-1
> Based on the listed generated examples in Fig.2, most faces are frontal, especially along Z axis, not sure if the variation of Z can determine the head orientation.
> Answer 2-1
> To convince the reviewer#2 that the latent vector z has a control over not only the head orientation but also the other parts that are not directly correlated to the identity of a speaker, we attached additional supplementary images in Appendix D.
>
> Question2-2
>  Faces in the third row and the forth may not keep the identity that well, especially the comparison between the face at the first column with others.
> Answer 2-2
> Although the conditional vector “c” captures the identity, we do not think the identity has to be always consistent regardless of latent vector “z” and that is the crucial part of our work that we tried to address in the Introduction (paragraph3, first line: “We believe, however, that it is impossible to perfectly reconstruct all the attributes in the image of a person’s face through the characteristics of the voice alone”).
>
> Question 3
> One typo may need to be addressed in the first paragraph of page 5, there are “the the” before word ‘inference’.
> Answer 3
> Thank you for pointing out the typo. We fixed this in the revised manuscript.
>
> Question 4
> As listed in Table 2, it seems the proposed approach achieves better performance when using 10 way training compared to other frameworks, any more analysis  why the proposed framework can achieve better performance in 10 way but obtain less accuracy in 2 way settings?
> Answer 4
> We are also aware of this fact and we have to admit that it is hard to interpret this result very clearly. One possible reason might be the difference of the model architecture, especially the classifier parts.
> In the work of SVHF they used an additional classifier rather than comparing the distance between embeddings on top of the networks. We conjecture that this might burden the classifier when the K gets bigger.
> DIMNET trains the core parts of each speech encoder and face encoder network separately, and K-way setting is not explicitly reflected into the training procedure by means of relative comparison using softmax and cross entropy error.
> Our model, however, is trained using the softmax function and does not require an additional classifier network, which might explain the robustness on relatively big K.
>
> Question 5
> Does the test data of 2 way or 10 way experiments also include the same ratio of positive and negative pairs? If so, how about the performance on the same validation data?
> Answer 5
> I don’t think I understand your question well. Regardless of K, there is always one positive pair (so the ratio is 50:50 for a 2-way setting and 10:90 for a 10-way setting?)
>
> Question 6
> About the results listed in Table 2, does the number 35^2 indicate 0.35?
> Answer 6
> It indicates 35(%) and the superscript denotes the footnote number. We modified this in the revised manuscript as the superscript could confuse the readers.
>
> Question 7
> It may be necessary to include these conditional face video generation works in the related work.
> Answer 7
> We will try to include other multimodal generation works in Related Works section, such as [3].
>
> References
> [1] O. M. Parkhi, A. Vedaldi, and A. Zisserman. Deep face recognition. In British Machine Vision Conference, 2015.
> [2] Forrester Cole, David Belanger, Dilip Krishnan, Aaron Sarna, Inbar Mosseri, and William T Freeman. Synthesizing normalized faces from facial identity features. In CVPR, 2017.
> [3] E. Zakharov, A. Shysheya, E. Burkov, and V. Lempitsky, Few-shot adversarial learning of realistic neural talking head models, arXiv preprint arXiv:1905.08233, 2019.

---

### Official Review · AnonReviewer4 · 2019-11-02
**Official Blind Review #4**

**Rating:** 8

**Review:**

Summary -
In this work the authors propose a two-stage procedure for training a GAN which generates plausible faces conditioned on the raw waveform of a speech signal. In the first stage two embedding functions are trained, one taking as input a frame from a video (of a person speaking), the other taking as input the raw waveform of the same video's audio. These embeddings are trained to maximize the inner product of positively sampled embeddings (frame and audio from same video), and minimize the inner product of negatively sampled embeddings (frame and audio from different videos). In the second stage a GAN is trained where the input to the generator is the a random latent code (z), concatenated with the learned embedding of a speech signal from stage 1 (c). They also initialize the first half (\phi) of the discriminator using the face embedding network from stage 1, and propose a modification of the relativistic GAN loss to prevent \phi from losing the ability to produce face embeddings that have low inner product with the wrong speech embedding.

The authors explore the properties of their learned pipeline in a number of experiments. In 4.1 they compare their self-supervised speaker matching pipeline with prior work, showing that it has competitive performance with prior work that relies on either supervised pretraining, or additional labels such as age and gender. In particular they show that as the number of negative samples increases from 1 to 9, their identity matching performance significantly improves over prior work (in the K=2 regime their method underperforms prior work). Qualitatively they show that their generator has some reasonable success (it is hard to judge what perfect success would look like, and how far they are from reaching it) at disentangling aspects of facial appearence that can and cannot be inferred from the speech signal (QLA1). They also show qualitatively that the output of their generator can be smoothly controlled by interpolating between speech conditioning vectors, producing reasonable faces at each intermediary step (QLA2). In experiment QTA1 they quantitatively validate the conditioning vector c is affecting the generated image. In experiment QTA2 the authors (with a small issue, see weaknesses) show that their proposed modification to the loss function causes their outputs to be better matched to the speech conditioning as measured by the fixed embedding network trained in stage 1. Finally in QT3 they measure how well their generated faces can be used with the original face embedding network to perform image retrieval (I am a bit unclear on the details of this experiment, see questions)


Strengths -
* The authors' proposed pipeline and modified loss function offers a generalized framework for jumpstarting conditional GAN training
* Their pipeline does not assume human-specified labels, but instead access to paired data from different modalities, which is can easily be obtained for many conditional image generation tasks (inpainting, super-resolution, colorization, etc.)
* Their pipeline also doesn't seem particularly finetuned for the speech-driven image synthesis task, so it seems reasonable to believe it could be adapted to other tasks
* Their results are qualitatively compelling, and they make a convincing efforts in experiments QTA1-3 to quantitatively show that the conditioning information is affecting the output
* The paper is generally well-written and easy to follow

Weaknesses -
* Without completing the loop, and showing that the second stage of this pipeline helps with identity matching for speakers, I'm not clear on what the motivation for this particular form of conditional image generation is (this is unfortunate, because their framework is quite general, and it seems feasible that the authors could have applied it directly to a task where the output of conditional image generation is directly useful)
* I interpret the main purpose of experiment QTA 2 as validating the effectiveness of their proposed loss modification, it seems like a natural experiment to make this claim more convincing is to compare a generator trained with Eq 4. against real images (hopefully getting a much lower matching probability than the 76.65% reported in the first experiment of this section)
* It seems like an important ablation study is testing the effect of jumpstarting the GAN training with the pretrained networks of stage 1, and reporting what happens when one or both of these networks are initialized randomly (assuming that initializing randomly hurts performance significantly, this would be further evidence of their framework's strengths)
* The novelty of the proposed approach is limited so far as I can tell (slight architectural modifications, and adding a negative sampling term to the discriminator loss)

Initial Rating - Weak Accept

Questions -
* In experiment QTA3 I'm confused by how including multiple faces from the same video clip affects measuring retrieval accuracy. Does retrieving any of the 50 faces from the same clip count as correct?

Explanation of Rating - The novelty of the work is limited, and it doesn't seem clearly useful for any practical task. However the stated task is certainly a non-trivial one, and the qualitative results and experiments give compelling evidence that the authors are proposing a powerful framework for conditional image generation. I think that the high quality writing, experiments, and results; coupled with potential impact for other conditional image generation tasks warrant acceptance. However the paper is held back by the lack of novelty and lack of clear motivation.

Revised Rating After Rebuttal: Accept

Explanation of Revised Rating: The authors addressed my concrete concerns about missing experiments in the rebuttal. Despite my concerns about the motivation for this particular task, I think the good results produced by author's methodology indicate this work will be valuable in the context of conditional image generation more broadly. I am upgrading my rating to accept.


**Experience Assessment:**

I have read many papers in this area.

**Review Assessment: Checking Correctness Of Derivations And Theory:**

N/A

**Review Assessment: Checking Correctness Of Experiments:**

I carefully checked the experiments.

**Review Assessment: Thoroughness In Paper Reading:**

N/A

---

> ### Author Response · Authors · 2019-11-08
> **Responses to AnonReviewer4**
>
> Before answering the question, we thank AnonReviewer4 for the detailed and kind review.
> Below are the responses regarding your concerns and answers to the questions.
>
> Weakness 1.
> Without completing the loop, and showing that the second stage of this pipeline helps with identity matching for speakers, I'm not clear on what the motivation for this particular form of conditional image generation is (this is unfortunate, because their framework is quite general, and it seems feasible that the authors could have applied it directly to a task where the output of conditional image generation is directly useful)
>
> Response: We agree with the reviewer’s opinion that our framework can be extended and applied to other modalities and datasets. The initial motivation of our work, however, was to seek the possibility of generating human face from the given speech segment, which is an interesting subject in itself and for voice profiling studies. In the middle of this project, we also became aware that this framework could work for other scenarios but we decided to concentrate on this project in accordance with our initial motivation because applying it to various scenarios might deteriorate the initial motivation and aim of our work. As we stated in the Conclusion & FutureWork section we plan to apply our methods to other modalities or other possible datasets.
>
> Weakness 2.
> * I interpret the main purpose of experiment QTA 2 as validating the effectiveness of their proposed loss modification, it seems like a natural experiment to make this claim more convincing is to compare a generator trained with Eq 4. against real images (hopefully getting a much lower matching probability than the 76.65% reported in the first experiment of this section)
>
> Response: The main purpose of experiment QTA2 was to quantitatively show that the generated face is not random but faithfully reflects information in a given speech segment. However, it is true that the part of the experiment was to show the effectiveness of the proposed loss function. Therefore, we ran the experiment again and found that the model without relidGANs loss yielded a much lower 47.2% matching probability as expected. We reported this result in the revised manuscript.
>
> Weakness 3.
> * It seems like an important ablation study is testing the effect of jumpstarting the GAN training with the pretrained networks of stage 1, and reporting what happens when one or both of these networks are initialized randomly (assuming that initializing randomly hurts performance significantly, this would be further evidence of their framework's strengths)
>
> Response: We agree with the reviewer’s concern and we actually tried training the model by randomly initializing the speech encoder and discriminator. However, we found out that the generator ignores the condition vector “c” from speech encoder and the latent vector “z” dominates the output of the generator. This is somehow expected and it shows that the inference training stage is necessary for speech encoder to extract the useful information regarding the cross-modal representation.
> In addition to that, we ran another experiment by training the discriminator from scratch while using the trained speech encoder. Surprisingly, we found out that, the resulting images were not much worse than those generated using the trained discriminator. However, we observed that the training gets significantly faster by initializing the discriminator with the trained face encoder. For example, we were able to obtain promising results within one epoch when initializing the discriminator with the trained face encoder, whereas it took at least a few epochs to get comparable results with random initialization.
>
>
> Weakness 4.
> * The novelty of the proposed approach is limited so far as I can tell (slight architectural modifications, and adding a negative sampling term to the discriminator loss)
> Response: Limited as it may seem, we believe that the major novelty of our work comes from the proposed system itself, which seamlessly integrates the inference and generation stages. We also believe our approach may suggest some fruitful ideas and directions for other studies aiming for other generative tasks where the human-labeled data is scarce.
>
>
> Question 1
> * In experiment QTA3 I'm confused by how including multiple faces from the same video clip affects measuring retrieval accuracy. Does retrieving any of the 50 faces from the same clip count as correct?
> Answer 1
> Yes, and we followed this based on the experiment of Speech2Face. Therefore, the random chance for top1 retrieval becomes 50/5000 * 100 = 1%.

---

### Public Comment · ~Yandong_Wen1 · 2019-10-19
**Experimental settings**

Thank you for sharing this great work. The proposed method is novel, and the experiments are quite comprehensive.

I am particularly interested in the usage of the datasets. I think it would be helpful if you could provide more details.

1. For AVSpeech dataset, is there any identity overlapping between the training and testing part?
2. For the experiments in Table 2, how do you split the training/testing set on Voxceleb and VGGFace?
3. Correct me if I'm wrong. All the experiments in section 4.2 (including the qualitative and quantitative ones) are conducted on AVSpeech dataset.

Many thanks to your response.

---

> ### Author Response · Authors · 2019-10-19
> **Thank you for your interest in our work!**
>
> We thank you for your compliments and interest in our work.
> Below are the responses to your questions.
>
> Q1. For AVSpeech dataset, is there any identity overlapping between the training and testing part?
>
> A1. The AVSpeech dataset is publicly available and the testset is already split by the authors of the dataset. Since the dataset does not explicitly provide the speaker identity information, we cannot be 100% sure that there exist no duplicated speakers between the trainset and the testset.
> That being said, we found that it is really hard to find duplicated speakers not only between the trainset and the testset but even within the trainset as most of the clip segments contain one speaker identity of a random person on YouTube (not public figures) which is why the fully self-supervised learning could work with a random negative sampling strategy.
> To put it briefly, we think the chances are very low, that there exist duplicated speakers between the trainset and the testset of AVSpeech dataset.
> Note that the AVSpeech dataset was also used in the recent work of Oh et al. (http://openaccess.thecvf.com/content_CVPR_2019/html/Oh_Speech2Face_Learning_the_Face_Behind_a_Voice_CVPR_2019_paper.html)
>
>
>
> Q2. For the experiments in Table 2, how do you split the training/testing set on Voxceleb and VGGFace?
>
> A2. We simply split the dataset following the experiment settings of Nagrani et al (https://www.robots.ox.ac.uk/~vgg/publications/2018/Nagrani18a/nagrani18a.pdf).
> That is, the names that start with ‘A’ or ‘B’ were used as a validation set and the names that start with ‘C’, ‘D’, ‘E’ were used for testset. The rest of the speakers were used as the trainset.
> We will try to elaborate on this in the following modified manuscript.
>
>
>
> Q3. Correct me if I’m wrong. All the experiments in section 4.2 (including the qualitative and quantitative ones) are conducted on AVSpeech dataset.
>
> A3. You are right. All the experiments were conducted on the AVSpeech dataset.

---

### Author Response · Authors · 2019-11-09
**Revised paper uploaded.**

We would like to thank all the reviewers for their kind comments and thoughtful analysis that help make our paper more complete.
We have uploaded a newly revised paper reflecting the comments, concerns and suggestions.
The revised parts are as below,
1. Minor typos and inconsistent reference format are fixed in the revised version now.
2. We ran an additional experiment to address the concerns of AnonReviewer4 (QTA2) and is reflected in the revised version now.
3. We added some supplementary images in Appendix D to address the concerns of AnonReviewer2. We generated some images by interpolating two randomly sampled latent vector 'z's by fixing 'c' to show that 'z' is able to capture the image attributes that are not directly correlated to the identity of a person (e.g., head orientation, hair color, attire, and glasses).
4. We revised the Related Works section to address the concern of AnonReviewer2. We emphasized more on the difference between the Speech2Face and our work to address that our proposed framework does not need any human labeled annotation.

---

### Comment · Area_Chair1 · 2019-11-14
**Reviewers, any comments on the author responses?**

Dear Reviewers, thanks for your thoughtful input on this submission!  The authors have now responded to your comments.  Please be sure to go through their replies and revisions.  If you have additional feedback or questions, it would be great to get them this week while the authors still have the opportunity to respond/revise further.  Thanks!

---

### Decision · Program_Chairs · 2019-12-19

**Decision:**

Accept (Poster)

**Comment:**

The authors propose a conditional GAN-based approach for generating faces consistent with given input speech.  The technical novelty is not large, as the approach is mainly putting together existing ideas, but the application is a fairly new one and the experiments and results are convincing.  The approach might also have broader applicability beyond this task.